# Solubility and Thermal Stability of *Thermotoga maritima* MreB

**DOI:** 10.3390/ijms232416044

**Published:** 2022-12-16

**Authors:** Beáta Longauer, Emőke Bódis, András Lukács, Szilvia Barkó, Miklós Nyitrai

**Affiliations:** 1Department of Biophysics, Medical School, University of Pécs, Szigeti Str. 12, H-7624 Pécs, Hungary; 2MTA-PTE Nuclear-Mitochondrial Interactions Research Group, Szigeti Str. 12, H-7624 Pécs, Hungary; 3Szentágothai Research Center, University of Pécs, H-7622 Pécs, Hungary

**Keywords:** MreB, solubility, heat stability, spectroscopy, salt bridge, nucleotide

## Abstract

The basis of MreB research is the study of the MreB protein from the *Thermotoga maritima* species, since it was the first one whose crystal structure was described. Since MreB proteins from different bacterial species show different polymerisation properties in terms of nucleotide and salt dependence, we conducted our research in this direction. For this, we performed measurements based on tryptophan emission, which were supplemented with temperature-dependent and chemical denaturation experiments. The role of nucleotide binding was studied through the fluorescent analogue TNP-ATP. These experiments show that *Thermotoga maritima* MreB is stabilised in the presence of low salt buffer and ATP. In the course of our work, we developed a new expression and purification procedure that allows us to obtain a large amount of pure, functional protein.

## 1. Introduction

Eukaryotic actin, and its distant bacterial homolog MreB, are important for the maintenance of cell shape and functionality. MreB plays a critical role in the assembly and organisation of the bacterial cell wall. Therefore, it is a potential drug target to combat multidrug resistance in bacteria. MreBs have been studied in several bacteria. MreB from the hyperthermophilic *Thermotoga maritima* (*Tm*-MreB) has been particularly intensively studied [1,2] including the elucidation of the first 3D structures of the monomers and filaments [3]. MreBs of other species studied include those of *Bacillus subtilis* (*Bs*-MreB) [3,4] *Chlamydophila pneumoniae* (*Cp*-MreB) [5], *Escherichia coli* (*Ec*-MreB) [6], and *Leptospira interrogans* (*Li*-MreB) [7].

These studies have shown that there are differences in filament dynamics between MreB homologs in vitro, including polymerisation propensity upon addition of nucleotides and/or salt, timing of nucleotide hydrolysis before, during, or after the polymerisation, or whether there are discrete actin-like nucleation and elongation steps of polymerisation. MreBs can form a variety of supramolecular structures in vitro when assembled under different conditions [3,4,8].

There is a contradiction in the MreB literature regarding the role of nucleotides in the polymerisation of MreBs from different species [3,4,5,7]. It has been described that MreB proteins from different species are able to polymerize in the presence but also in the absence of nucleotides [3,6,8]. Therefore, it is still debatable whether MreB proteins require nucleotides to maintain normal function or not. On the other hand, since ATP is present in millimolar concentrations in cell cytoplasm [9], it can be assumed that MreB binds nucleotides under cellular conditions. Moreover, it was recently described that ATP binding increases the melting temperature of Sc-MreB5 [10].

The optimal purification method and storage conditions for MreB protein are also questionable. The MreB laboratories are divided into two groups, namely the salt-free [4,5,11] and high-salt preferring [6,8,12]. The presence of ATP also plays a central role in these experiments. In salt-free, ATP containing methods a monovalent salt is used to initiate polymerisation, while the high-salt buffer requires ATP to induce the assembly of monomers.

Our aim was to describe the thermal stability of *Tm*-MreB in relation to the presence of nucleotide and salt [11]. To this end we took advantage the spectroscopic signal from a single tryptophan of *Tm*-MreB. As described previously, the local conformational changes of the tryptophan residue can be used to detect thermal denaturation [13,14]. On this basis, we found that conformation and stability of *Tm*-MreB strongly depend on the buffer conditions. The absence of monovalent ions and the binding of ATP favour the most thermostable form, although the solubility is increased in a salt-rich environment.

In our previous studies, we found that millimolar magnesium and one hundred millimoles of potassium or sodium are required for efficient MreB polymerisation. The addition of calcium causes arrangement in ribbon-like structures and large bundles, and we hypothesise that calcium binding may alter the filaments [15]. We assume that the polymerisation inhibiting effect of salt (NaCl or KCl) can be explained by the less stable conformation of MreB, which affects the salt bridge between two subdomains of *Tm*-MreB.

We also described the ATP binding ability of *Tm*-MreB and found that it is higher under salt-free conditions. It was also described that the widely used, potential antibiotic MreB-specific inhibitor A22 [16] does not affect ATP binding under any conditions.

## 2. Results

In this paper, we describe a novel native purification method that enables the production of soluble, functional MreB protein in high purity and quantity.

The first and probably most important point is that, in our recent work, we have used for the first time a cell line containing endogenously expressed chaperone proteins. These help to maintain a native conformation of the expressed protein during expression. Our previous experience, and also other publications, have shown that MreB proteins can be produced in good quantity in recombinant expression systems, but the yield is poor. This is because most of expressed protein is lost in the first step of ultracentrifugation and subsequent cell lysis. This can probably be explained by the formation of “inclusion bodies”, i.e., unfolded protein aggregates.

When using ArcticExpress (DE3) competent cells, the expression of MreB is high (Figure 1A, line 2) and, furthermore, there is almost no loss upon ultracentrifugation; the full amount of MreB is found in the supernatant (Figure 1B, lines 3 and 4).

Following the protocol described by van den Ent et al. in 2000 [3], the Tm-MreB can be eluted using an increased imidazole concentration of Ni resin (Figure 1B, lanes 5, 6 and 7). The histidine tag can be removed (Figure 1C, lanes 9 and 10), and further concentration of the protein is even possible without aggregation (Figure 1C, lane 11).

It is worth noting that this step has been found to be crucial for the purification of MreB, i.e., the buffer conditions play a crucial role in determining the structure and solubility of MreB. In our experiments, we used two different buffer conditions that are most popular in the MreB literature. The first was originally described in the publication mentioned above [3] (20 mM TRIS-HCl, 200 mM NaCl, 1 mM EDTA, pH 7.5) The second is the one which is referred to in the “actin literature” [4] (4 mM TRIS, 0.1 mM CaCl_2_, with or without 2 mM ATP, pH 7.5) and is commonly referred as buffer A. For better understanding, the first buffer is indicated as high-salt, and the second as salt-free in this paper.

It seems that a high-salt buffer is more suitable for MreB. At high-salt concentration, imidazole subtraction can be performed without aggregation using the PD10 desalting column. In buffer A hard, visible precipitation of MreB was observed in the absence of ATP, but some aggregation of MreB was also seen in the presence of ATP.

Interestingly, *Tm*-MreB does not precipitate when buffer conditions change from high-salt to salt-free. Therefore, to study MreB in a salt-free environment, a fraction of *Tm*-MreB was dialyzed in buffer A.

To describe the difference in stability in different buffer conditions, MreB was investigated by a spectroscopic method. *Tm*-MreB contains a tryptophan residue that allows us to describe conformational changes under different conditions.

We used a temperature-controlled cuvette holder that allowed fine tuning of the temperature in the cuvette and simultaneous observation of the tryptophan signal with a spectrofluorometer. Our aim was to determine the thermal unfolding of *Tm*-MreB and subsequently measure the decrease in tryptophan fluorescence, as has been described for other proteins [17,18].

First, it is worth noting that the tryptophan residue of *Tm*-MreB is probably accessible to the solvent since the maximum of fluorescence emission is by 350 nm [19]. In our experiments, the fluorescence emission of *Tm*-MreB decreased with increasing temperature (Appendix A). Interestingly, the fluorescence emission decreased significantly upon thermal denaturation (Appendix A, lowest, orange line).

At high-salt conditions *Tm*-MreB does not appear to be stable, as tryptophan fluorescence intensity decreases with increasing temperature, up to the point of final denaturation at 50 °C (Figure 2). This is in good agreement with previous data described by CD spectroscopy (Table 1) [11]. Our data show that the presence of ATP at high concentration (2 mM) can stabilise *Tm*-MreB, the denaturation is shifted to 55–60 °C.

Surprisingly, our data showed that *Tm*-MreB in buffer A exhibits higher thermal stability than in a salt-containing environment. In a salt-free environment, *Tm*-MreB shows much higher heat stability: it is evident from the spectroscopic data (and visually) that it remains stable until 75 °C (Figure 2). The most surprising data, however, is that *Tm*-MreB does not denature in the presence of 2 mM ATP, at least to 90 °C (Table 1).

It can be concluded that the preferred method is a two-step purification of MreB; imidazole removal should be done in a buffer with high-salt content, but the desirable storage buffer is a salt-free buffer such as buffer A. With this purification method, high protein purification efficiency can be achieved: about 30–40 mg of purified MreB can be obtained from 3 g of cell pellet.

It is worth mentioning that we did not observe any wavelength shift of the tryptophan residue when heated, but that the fluorescence intensity decreased with increasing temperature (Appendix A). This observation is fundamentally different from that of actin (Appendix A).

To describe chemical denaturation characteristic of *Tm*-MreB, denaturation experiments were performed with guanidine hydrochloride. In this case, a decrease in fluorescence emission can also be observed, as previously described [20]. Our data show that in these experiments the emission maximum of the tryptophan residue was shifted towards to higher wavelength (Appendix A). This behaviour was observed in both buffer conditions (Figure 3).

It was also investigated which parameters influence the quantum yield of the tryptophan residue. First, tryptophan residue was investigated as such in different buffer conditions. Our data showed that the fluorescence intensity is independent of the buffer conditions (Figure 4, dashed lines). On the other hand, the tryptophan residue of *Tm*-MreB was found to have about four-to-ten times lower fluorescence intensity as tryptophan residue in solution (Figure 4, straight lines). This can probably be explained by the shadowing effect of the local environment. Furthermore, the fluorescence intensity of the tryptophan residue in a salt-free environment is much lower (blue curve) than in the presence of salt (red curve). This can be explained by the closed conformation near the tryptophan residue compared to a salt-rich environment. Interestingly, the binding of ATP has no influence on the emission of the tryptophan signal (Figure inset).

Based on previous data, we aimed to characterise ATP binding to *Tm*-MreB in this work. In our experiments, we used a fluorescent, non-hydrolysing ATP analogue, TNP-ATP, which shows an increased fluorescence signal upon protein binding [21].

When plotting the fluorescence signal of TNP-ATP as a function of *Tm*-MreB concentration in buffer A, saturation can be observed (Figure 5). In the case of saturation, the bound fraction reaches its maximum so that a Kd value can be determined at half saturation.

Our data show that the affinity of TNP-ATP for *Tm*-MreB in salt-free environment is 3.69 ± 0.04 µM. This is a relatively weak binding, which is probably higher under in vivo conditions due to the binding of other factors to MreB.

Interestingly, our data showed that, although TNP-ATP binds to *Tm*-MreB in both buffer conditions, the binding in buffer A is stronger than in the presence of salt. The fluorescence intensity of TNP-ATP is about twice as high at the same concentration of *Tm*-MreB (Figure 6). We also investigated whether the MreB-specific inhibitor A22 can block ATP binding to *Tm*-MreB. Our spectroscopic data clearly show that A22 has no effect on TNP-ATP binding at all buffer conditions applied.

To describe whether TNP-ATP binds to the same region of *Tm*-MreB as ATP, competition assays were performed. Our data show that ATP can displace TNP-ATP, i.e., the fluorescence signal of TNP-ATP decreases upon addition of ATP (Figure 7). This suggests that TNP-ATP binds to the nucleotide binding pocket of MreB and is a competitor of ATP. Based on previous publications, we were able to determine the affinity of ATP for *Tm*-MreB [22], which was about 2 μM.

## 3. Discussion

Our data show that *Thermotoga maritima* MreB can be expressed and purified in large quantities using ArcticExpress (DE3) competent cells. The chaperones of this system can help produce native, soluble *Tm*-MreB so that it does not form inclusion bodies. Therefore, native purification can be performed instead of denaturation as previously described [7]. The most sensitive step in purification is the removal of the imidazole, probably because imidazole can stabilise MreB. We have described that this step is crucial and requires a high-salt concentration to obtain a soluble protein, i.e., in case of direct removal of imidazole in buffer, precipitation of MreB can be observed.

Apart from this, these buffer conditions are not particularly suitable for prolonged storage of *Tm*-MreB. We have observed that *Tm*-MreB has a very low melting temperature in a high-salt environment, especially in the absence of ATP. The presence of ATP can increase the stability of *Tm*-MreB, but it is still very low compared to salt-free conditions. It can be concluded that *Tm*-MreB has a significantly increased thermal stability in a salt-free environment, i.e., in presence of TRIS-HCl and the divalent cation CaCl_2,_ it is around 75 °C. Moreover, the presence of ATP may further stabilise *Tm*-MreB, as it exhibits a stable conformation above 90 °C. This is in good agreement with previous observations, even though a much lower ATP concentration was used in these experiments [11] The extremely high melting temperature can be explained by the habitat of *Thermotoga maritima*, which lives in hot springs, requiring extreme heat stability. In summary, *Tm*-MreB belongs to the group of proteins that require ATP for stable conformation [23].

The possible explanation for the difference in heat stability in salt-free and high-salt conditions is the presence of a salt bridge in the molecule. It has already been described that salt bridges, which can form between oppositely charged residues, such as Arg € or Lys (K) and Asp (D) or G€ (E), can stabilise or even destabilise the structure of proteins [24]. The enhancement of thermal stability by salt bridges has already been described for other thermophilic proteins [25], and ATP has been found to have a concentration-dependent effect on the heat stability of a large number of proteins [23]. Salt bridge abolition by mutation and higher ionic strength weakens the salt bridge and destabilises certain proteins [26].

In the case of *Thermotoga maritima* MreB it is known that there is a salt bridge between K49 and E204 [3] (Figure 8). It is worth noting that the two residues localise on IB and IIB subdomains of *Tm*-MreB (Figure 8). It has already been described that the spatial orientation of these subdomains with respect to each other is closely correlated with the binding of ATP in MreB [27]. Therefore, this “flattening” motion may play a central role in changing the conformation during polymerisation, as is the case with actin [27]. We conclude that salt bridge formation is more favourable in the flattened conformation because the amino acids that form the salt bridge are much closer in this conformation. It has been described that a high-salt concentration can increase the solubility of the protein, which is referred to as “salting in” the process [28]. On the other hand, the presence of monovalent ions weakens the salt bridge [29], and thus affects thermal stability [30]. Our hypothesis is that NaCl stabilises *Tm*-MreB structure during cell lysis and the initial purification steps, but inhibits the formation of the salt bridge. As the stable, soluble structure is evolved, removal of NaCl allows the salt bridge to form and thus increases the thermal stability of *Tm*-MreB, especially when ATP is bound in the nucleotide binding pocket [23].

We have described that the quantum yield of fluorescence of tryptophan residue decreases with increasing temperature, which can be explained by the interaction of chromophores with quenching agents in the solvent or in the protein itself. On the other hand, almost no shift in fluorescence emission from the tryptophan residue was observed during thermal unfolding, although it can be observed in actin (see Supplementary data). Normally, the theory is that the maximum of the fluorescence emission undergoes a red shift when the chromophores are more exposed to the solvent. In our experiment, this was not the case, regardless of the buffer conditions or the nucleotides.

To describe the behaviour of the single tryptophan of *Tm*-MreB upon denaturation, a chemical denaturation assay was performed. Interestingly, a typical red shift was observed in this case. From this, we can conclude that thermal and chemical denaturation induce different conformational changes on *Tm*-MreB and that, in the latter case, the environment of the tryptophan is more exposed to the solvent. On the other hand, it is worth noting that actin contains four tryptophan amino acids and therefore the fluorescence intensity of actin is determined by the combination of these amino acids (Figure 9).

Our spectroscopic data show that tryptophan is buried in the molecule, and this shadow effect is more pronounced in salt-free environments. In agreement with the thermal denaturation data, this may indicate a more stable conformation.

It was described earlier that the four tryptophan of eukaryotic actin contributes differently in the intrinsic fluorescence intensity of actin [31]. It was found that Trp-79 is the most accessible for the solvent; therefore, it is the most sensitive to the changing of buffer conditions. As it can be seen on the Figure 9, the only tryptophan residue of *Thermotoga maritima* MreB is in a closed environment; therefore, we understand that it is not easily affected by environmental changes, i.e., ion conditions. This can explain why there was no red shift of tryptophan emission. On the other side, it is under the nucleotide binding cleft, between the two subdomains of *Tm*-MreB; therefore, its quantum yield can be sensitive to conformational changes of the molecule.

The non-hydrolysing ATP analogue TNP-ATP proved a suitable agent to describe nucleotide binding of *Tm*-MreB. Supporting our thermal denaturing assays, where the stable conformation of *Tm*-MreB was found via a salt-free environment, it was also found that, in salt-free circumstances, *Tm*-MreB binds TNP-ATP more tightly than in the presence of monovalent ions at high concentration. On the other side, it was found that the MreB specific inhibitor A22 does not inhibit binding of TNP-ATP to MreB in any circumstances. Finally, we concluded that ATP is a competitor of TNP-ATP, which means that they have the same binding region. It was found that approximately 35–40% of bound TNP-ATP could be removed by ATP; therefore, it can be suggested that there are binding sites with different affinities in MreB. The 2 µM affinity of ATP to *Tm*-MreB shows a relatively weak binding; it can be concluded that other factors or binding partners in cytoplasm can improve that.

## 4. Materials and Methods

### 4.1. Protein Expression and Purification

#### 4.1.1. Expression

The plasmid containing Thermotoga maritima MreB was transformed into ArcticExpress (DE3) (Agilent Technologies, Santa Clara, CA, USA) competent cells. We added 2 µL of ß-mercaptoethanol diluted with distilled water at a ratio of 1:10 to 100 µL of competent cells. The cells were incubated on ice for 10 min, rotating the tubes by hand every 2 min. After 10 min, 5 µL of plasmid DNA was added to the competent cells. Transformed cells were then incubated on ice for 30 min. After a 20 s heat shock (42 °C water bath), the cells were placed on ice again for 2 min. The transformed bacterial cells were grown in 0.9 mL of pre-warmed Luria broth (LB) nutrient solution at 37 °C for 1 h, shaking at 220–250 rpm. Then 150 µL of cell culture was spread on a plate containing double resistance (gentamicin: 20 µg/mL and ampicillin: 100 µg/mL).

An independent colony was inoculated with 100 mL of LB solution containing the appropriate amount of both antibiotics. The cell culture was grown overnight in a shaking incubator at 37 °C (220–250 °rpm). The next day, 20-20 mL of cell culture was added to 1-1 L of LB nutrient solution and were shaken at 30 °C at 220–250 rpm for 3 h. Here, it is no longer necessary to add antibiotics to the cell culture. After the incubation period, the temperature was set to 11.5 °C. As the cells cooled back to 20 °C, 0.8 mM IPTG was added as a final concentration and the cultures were incubated for 24 h. The cells were centrifuged (2900× *g*, 10 min.) and the pellet was stored at −20 °C.

#### 4.1.2. Purification

The Thermotoga MreB cell pellet (1–2 g) was homogenized in TRIS-HCl buffer (50 mM TRIS, pH 8.0, 1 g pellet/10 mL buffer). After adding lysozyme, the cells were sonicated on ice (80%, pulse for 1 min, then 1 min break, repeated 6 times). Before centrifugation (328,000× *g*, at 4 °C, 30 min), DNase I was added to the lysate (50 µg/mL). The Ni-nitrilotriacetic acid (Ni-NTA) (Qiagen, Hilden, Germany) column was eluted with 5% imidazole buffer (1M imidazole, 50 mM TRIS-HCl, pH 6.0) and 95% NaCl buffer (300 mM NaCl, 50 mM TRIS-HCl, pH 6.0), equilibrated and the supernatant was cooled and stirred for 1 h. The column was then washed with 10–20–30–50% imidazole buffer (dissolved in NaCl buffer). The fractions were collected separately and analysed by SDS gel electrophoresis.

Fractions containing MreB were dialysed in high-salt buffer (20 mM TRIS-HCl, 200 mM NaCl, 1 mM EDTA, pH 7.5). One part of this was further dialysed against salt-free buffer (4 mM TRIS-HCl, 0.1 mM CaCl_2_, pH 7.5) with two buffer exchanges. *Tm*-MreB was ultracentrifuged (100,000× *g*, 4 °C, 30 min). The His-tag was removed with PreScission protease (GE Healthcare Life Sciences) (2U/100 µg protein) overnight at 4 °C. The protease was removed on GST column next day. The final protein concentration was measured with a spectrophotometer.

#### 4.1.3. Tryptophan Measurement

We measured 20 µM of *Tm*-MreB or 20 µM of tryptophan in salt-free (4 mM TRIS-HCl, 0.1 mM CaCl_2_, pH 7.5) or high-salt (20 mM TRIS-HCl, 200 mM NaCl, 1 mM EDTA, pH 7.5) conditions, using a Horiba Jobin Yvon spectrofluorometer. The excitation was set to 295 nm. The emission was measured between 310 nm and 450 nm at 22 °C.

#### 4.1.4. Thermal Denaturation

We heated 20 µM of *Tm*-MreB protein in different buffer conditions (salt-free (4 mM TRIS-HCl, 0.1 mM CaCl_2_, pH 7.5) or high-salt (20 mM TRIS-HCl, 200 mM NaCl, 1 mM EDTA, pH 7.5) in the presence or absence of 2 mM ATP) between 20–95 °C and the fluorescence intensity of tryptophan was continuously measured between 310 nm and 450 nm. The excitation wavelength was 295 nm. The measurements were performed on a Jobin Yvon Horiba fluorimeter equipped with a Quantum Northwest TLC50 temperature-controlled cuvette holder.

#### 4.1.5. Chemical Denaturation

We measured 20 µM of *Tm*-MreB by Horiba Jobin Yvon fluorimeter. The chemical denaturation was performed with guanidine hydrochloride solution (6M Gu-HCl was dissolved in salt-free or high-salt buffer). The guanidine was added to the protein in increasing concentrations in 0.2 M steps. The excitation was set to 295 nm and the tryptophan emission was detected by each step between 310 nm and 450 nm, at room temperature.

#### 4.1.6. TNP-ATP Binding to Tm-MreB

We incubated 20 µM of overnight in a salt-free (4 mM TRIS-HCl, 0.1 mM CaCl_2_, pH 7.5) or a high-salt (20 mM TRIS-HCl, 200 mM NaCl, 1 mM EDTA, pH 7.5) environment with 1 µM TNP-ATP in the absence or presence of 50 µM A22. The next day, the fluorescence intensity of TNP-ATP was measured. The excitation was set to 400 nm and the emission was measured between 420 nm and 650 nm. The maximal intensity of spectra (by 540 nm) was applied to calculate the binding ratio and was set to bound fraction 0 by no binding and 1 by saturation.

The Hill equation was fitted to curve:$$y=V_{max}*\frac{x^{n}}{k^{n}+x^{n}}$$
where *V_max_* is the maximal *y* data, *k* is *x* by half saturation and *n* is the Hill coefficient.

Kd of TNP-ATP to *Tm*-MreB was obtained *k*.

#### 4.1.7. ATP Competition Assay

We incubated 20 µM of *Tm*-MreB overnight in a salt-free (4 mM TRIS-HCl, 0.1 mM CaCl_2_, pH 7.5) environment with 1 µM TNP-ATP. On the next day, non-fluorescence ATP (pH 7.5) was added to samples and the fluorescence signal of TNP-ATP was measured.

Affinity of ATP to *Tm*-MreB was calculated using the Langmuir single-site binding equation for a curve fit based on previous publications [29].

## 5. Conclusions

We have described that Thermotoga maritima MreB can be expressed and purified in high amounts using ArcticExpress (DE3) competent cells. The most sensitive step during the purification of *Tm*-MreB is the imidazole removal because *Tm*-MreB remains soluble in the presence of high amounts of NaCl. By further storage and investigation, a salt-free environment and the presence of ATP improves the stability of *Tm*-MreB. Our data showed that MreB binds ATP with micromolar affinity and that MreB specific inhibitor A22 does not block the binding of ATP.

## Figures and Tables

**Figure 1 ijms-23-16044-f001:**
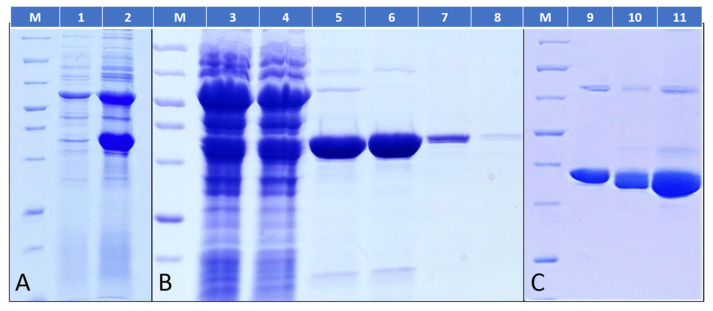
Purification of *Tm*-MreB using BL21 DE3 Arctic express cell line. (**A**) Molecular weight marker (M) followed by the cell lysates before (lane 1) and after induction with IPTG (lane 2). (**B**) Purification of *Tm*-MreB using high-salt purification buffers: lane 3: cell lysate before ultracentrifugation; lane 4: supernatant after ultracentrifugation; lanes 5–8: elution of protein with increasing imidazole concentration. (**C**) Last steps of the *Tm*-MreB purification: lane 9: *Tm*-MreB fraction before His-tag removal; lane 10: His-tag cleaved *Tm*-MreB; lane 11: concentrated, His-tag-free *Tm*-MreB.

**Figure 2 ijms-23-16044-f002:**
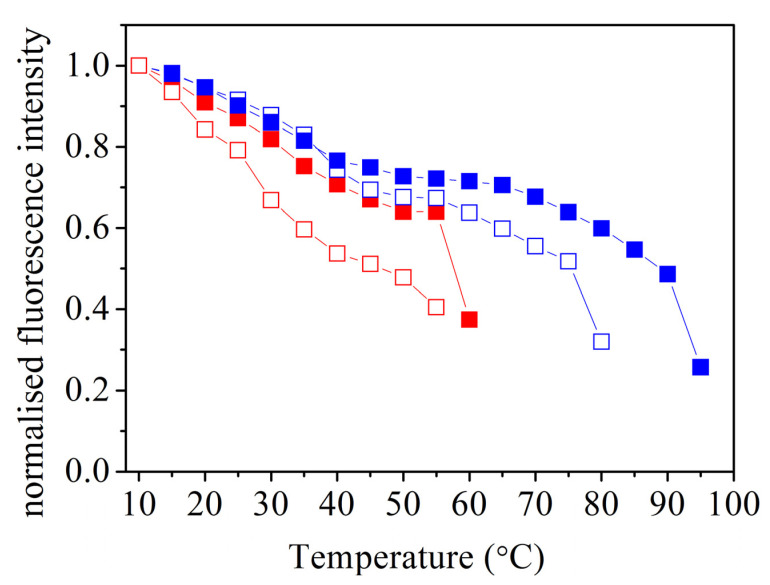
Thermal denaturation of *Tm*-MreB at different buffer conditions. The tryptophan fluorescence intensity of *Tm*-MreB was examined in high-salt (red) or salt-free (blue) environments. In one group of experiments, 2 mM ATP was added to MreB (closed symbols).

**Figure 3 ijms-23-16044-f003:**
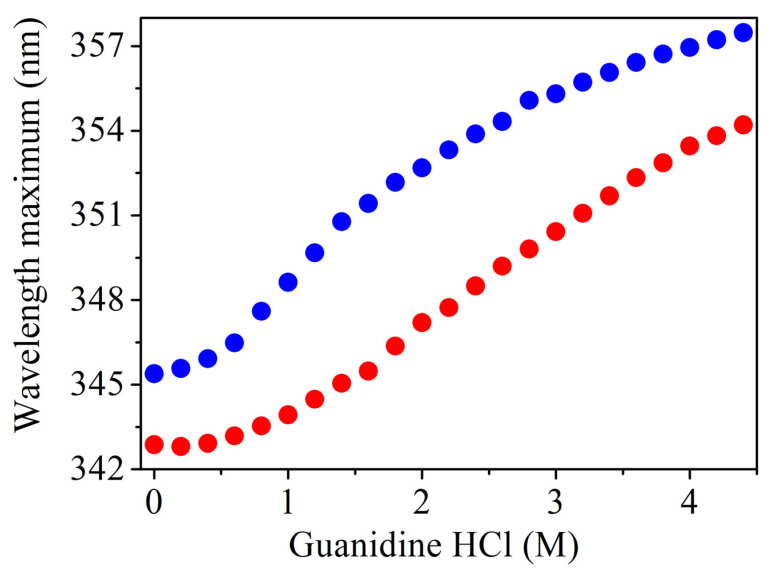
Chemical denaturation of *Tm*-MreB with guanidine hydrochloride. *Tm*-MreB can be denatured in high-salt (red) and even in salt-free (blue) environments. The emission spectra were fitted with the Gaussian function and the centre plotted as a function of the Gu-HCl concentration.

**Figure 4 ijms-23-16044-f004:**
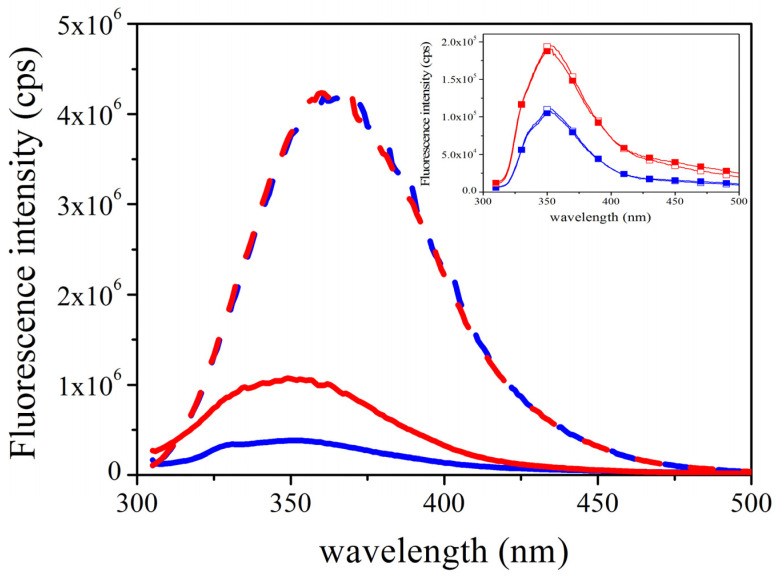
The tryptophan fluorescence intensity of *Tm*-MreB depends on the buffer conditions. The fluorescence intensity of the tryptophan residue was measured in salt-free (blue line) and high-salt (red line) environments. The fluorescence intensity of the tryptophan residue itself is not affected by buffer conditions (dashed lines). The tryptophan residue of *Tm*-MreB shows a lower fluorescence intensity than tryptophan alone at the same concentration, showing the shadowing effect of the surrounding residues in *Tm*-MreB. This effect is more pronounced in salt-free buffer (blue curve), probably due to conformational changes. **Inset:** The presence or binding of ATP has no effect on the fluorescence of tryptophan. In an independent measurement, the fluorescence intensity of *Tm*-MreB was measured in a salt-free environment (blue curves) and in the presence of salt (red curves) in the presence (filled symbols) and absence (empty symbols) of ATP.

**Figure 5 ijms-23-16044-f005:**
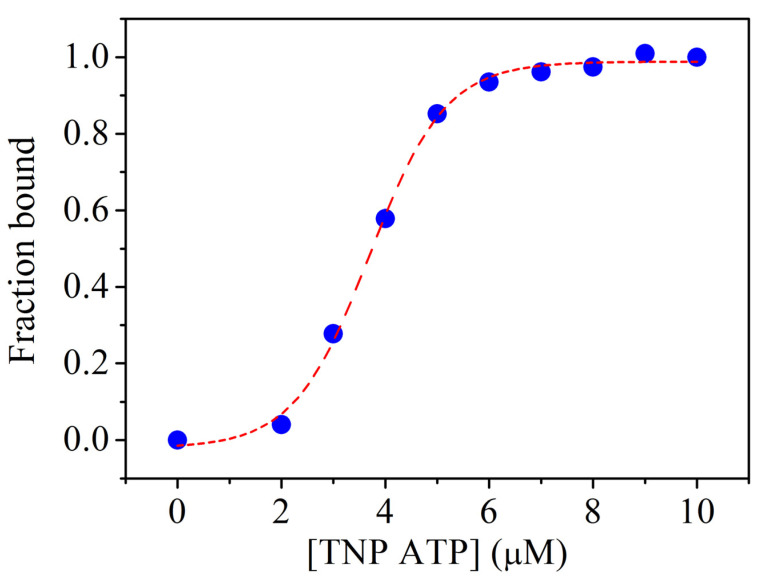
Binding of TNP-ATP to *Tm*-MreB shows saturation. The fluorescence intensity of TNP-ATP was measured in the presence of 20 μM *Tm*-MreB at different concentrations. In the case of saturation, no more binding is possible, so it was plotted as the maximum of the bound fraction. The affinity of *Tm*-MreB for TNP-ATP was calculated as half-maximum of saturation and determined to be 3.69 μM.

**Figure 6 ijms-23-16044-f006:**
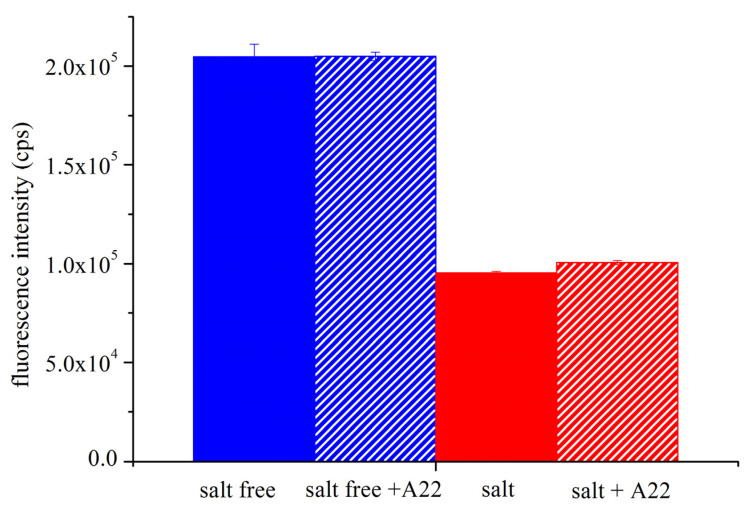
The fluorescent ATP analogue TNP-ATP binds more tightly to *Tm*-MreB in salt-free environment. We incubated 20 μM *Tm*-MreB overnight in a salt-free (blue) or high-salt (red) environment with 1 μM TNP-ATP, in the absence or presence of 50 μM A22 (+A22 indicated). The next day, the fluorescence intensity was measured from TNP-ATP.

**Figure 7 ijms-23-16044-f007:**
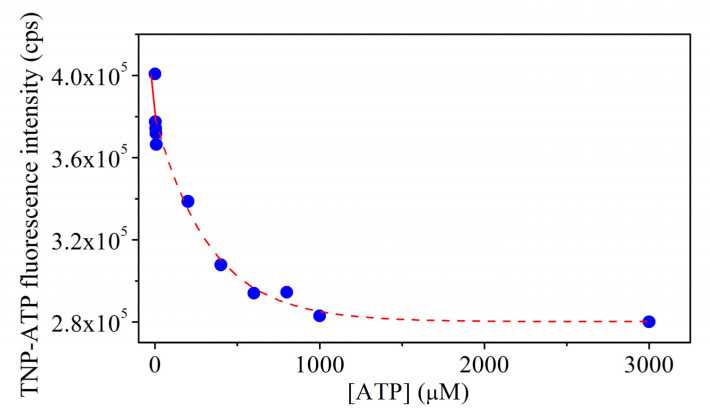
ATP is competitor of TNP-ATP. The fluorescence emission of TNP-ATP was measured in the presence of an increasing concentration of ATP. Based on the curve, the dissociation constant of ATP to *Tm*-MreB is about 3.7 μM.

**Figure 8 ijms-23-16044-f008:**
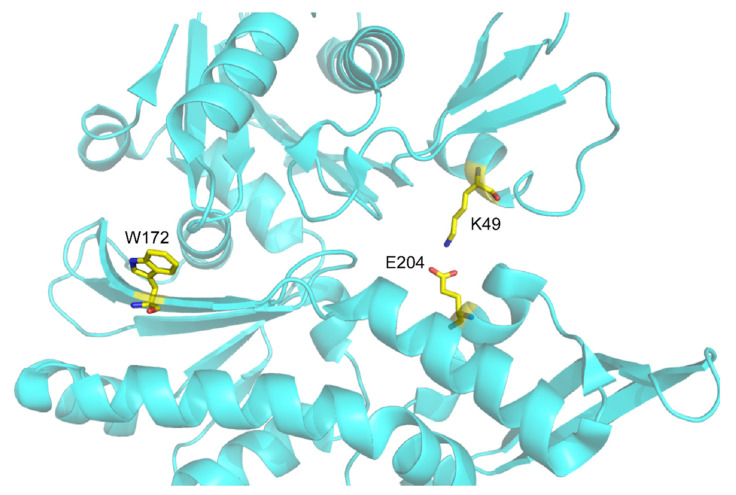
The salt bridge of *Tm*-MreB can affect the conformation of *Tm*-MreB and consequently the fluorescence emission of the tryptophan residue. *Tm*-MreB (PDB: 1JCG) K49 lysine and E204 glutamic acids are located on different subdomains; therefore, the salt bridge between them may affect the local environment of W172.

**Figure 9 ijms-23-16044-f009:**
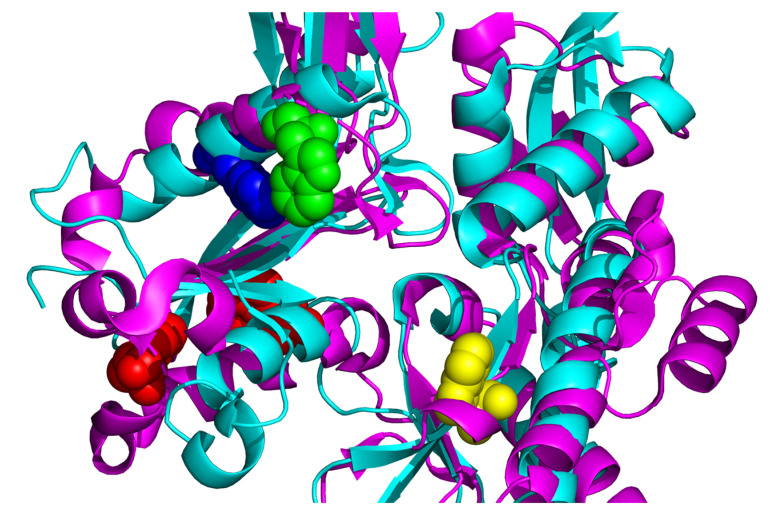
Localization of tryptophan residues in Tm-MreB and skeletal alpha-actin. Monomeric rabbit skeletal alpha actin (PDB: 1NWK) is in magenta; *Tm*-MreB (PDB: 1JCG) is in blue. The Trp-79 tryptophan residue of actin can be seen as green spheres; it is the most accessible for the solvent compared with the other three tryptophan residues of actin. The Trp-86 of actin, which is mostly quenched by cysteine and methionine residues, is in blue, and the most intrinsic residues are in red. The position of single tryptophan of *Tm*-MreB is in yellow; it is positioning separately from the tryptophan residues of actin. It does not seem as accessible to solvent as Trp-79 of actin, but it is in the cleft between two subdomains; therefore, it can be sensitive for conformational changes.

**Table 1 ijms-23-16044-t001:** Results of thermal denaturation of *Tm*-MreB. Second-order differentiation was performed using the data in Figure 2 to obtain the decay of the fluorescence signal, i.e., melting temperature.

	Melting Temperature	
Buffer Conditions	Previous Data (CD)	This Work (Tryptophan Signal)
Salt-free	N/A	75 °C
Salt-free + ATP	70 °C [11]	90 °C≤
High-salt	53 °C [11]	~50 °C
High-salt + ATP	N/A	55 °C

## Data Availability

Not applicable.

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
