# Peer review of "Solubility and Thermal Stability of *Thermotoga maritima* MreB"

_ijms, 2022, doi:10.3390/ijms232416044_

Round 1

Reviewer 1 Report

In this manuscript, the authors reported tryptophan emission-based experiments about Thermotoga maritima MreB and demonstrated that Tm-MreB can be stabilized in the environment which contains low salt and ATP. The manuscript is well written, though the authors need to address some concerns before it is considered for publication.

1. Line 20, references need to be added in the first three sentences.

2. Line 27, an extra space before hyphen in Cp-MreB’

Line 28, an extra space before hyphen in Li-MreB’

Line 44, an extra space before hyphen inTm-MreB’

3. Lin 34, the sequence of references should be [2][3][9].

    Line 38, the sequence of references should be [4][6][9][11].

3. Line 43, why was only A22 selected for investigating the spectral characteristics of Tm-MreB instead of other inhibitors?

4. Line 62, the quotation mark before inclusion bodies should be superscripted.

5. Line 268, the font format of “Localization of tryptophan residues in Tm-MreB and skeletal alpha-actin” is different from the rest of caption.

6. Line 406, the title of ref 1 should not be all upper case. “T.” should be Thermotoga.

Line 426, Pearl Nurse is author name, not title.

Please doublecheck other references and make sure they are using the   right format.

Author Response

Dear Reviewer,

Thank you for reaching out and providing us with valuable feedback. We would like to thank for your thoughtful comments and efforts towards improving our manuscript.

We are addressing comments specific to each statement as attached.

Reviewer 2 Report

The manuscript by Longauer et al conveys a biophysical study of the MreB protein from Thermotoga maritima, which is also expressed and purified in a new way by the authors. Fluorescence spectroscopy is used to follow thermal and chemical denaturation of the protein, and the binding of an ATP analogue, TNP-ATP, in different conditions. Results indicate a protein stabilization in low salt conditions, and upon ATP binding.

The paper does not clarify the ultimate goal of the biophysical characterization of MreB. Indeed, there is no connection between the experiments and the final conclusions regarding the protein and its function. The importance of MreB is commented in the beginning of the introduction, but the reason why the studies are performed is not well-defined. Experiments are well conducted, but nowadays using just fluorescence in biophysics might not be enough since usually additional complementary information is required. For instance, it seems that polymerization of MreB is important, but no size particle technique (such as DLS, SAXS, etc) is done. Finally, another flaw of the paper is why it is stated that the salt bridge cited in the discussion is the main reason for the results. Where is the conclusion provided from? Molecular dynamics, and other kind of experiments, would give other multiple reasons that may be the causes. Just fluorescence done with and without salts simply cannot explain why a specific salt bridge is the responsible for the stability when salts are present. Additionally, English language needs a really thorough revision. In conclusion, in my opinion, the paper should be rejected.

Author Response

Dear Reviewer,

Thank you for reaching out and providing us with valuable feedback. We would like to thank for your thoughtful comments and efforts towards improving our manuscript.

We are addressing comments specific to each statement below.

Round 2

Reviewer 2 Report

I think the authors have done a great effort to improve the quality of the manuscript, and therefore it deserves publication. However, both the issue of considering just the salt bridge as the cause of their results, and the use of more biophysical techniques, are two aspects that the authors should consider in their future/ongoing research.